# Methylglyoxal-Derived Advanced Glycation End Product (AGE4)-Induced Apoptosis Leads to Mitochondrial Dysfunction and Endoplasmic Reticulum Stress through the RAGE/JNK Pathway in Kidney Cells

**DOI:** 10.3390/ijms22126530

**Published:** 2021-06-18

**Authors:** So-Ra Jeong, Kwang-Won Lee

**Affiliations:** Department of Biotechnology, College of Life Sciences and Biotechnology, Korea University, Anam-Dong, Sungbuk-Gu, Seoul 02841, Korea; wjdxmfhf@naver.com

**Keywords:** methylglyoxal-derived AGEs, kidney injury, endoplasmic reticulum stress, mitochondrial dysfunction, JNK signal pathway

## Abstract

Advanced glycation end products (AGEs) are formed via nonenzymatic reactions between reducing sugars and proteins. Recent studies have shown that methylglyoxal, a potent precursor for AGEs, causes a variety of biological dysfunctions, including diabetes, inflammation, renal failure, and cancer. However, little is known about the function of methylglyoxal-derived AGEs (AGE4) in kidney cells. Therefore, we verified the expression of endoplasmic reticulum (ER) stress-related genes and apoptosis markers to determine the effects of AGE4 on human proximal epithelial cells (HK-2). Moreover, our results showed that AGE4 induced the expression of apoptosis markers, such as Bax, p53, and kidney injury molecule-1, but downregulated Bcl-2 and cyclin D1 levels. AGE4 also promoted the expression of NF-κB, serving as a transcription factor, and the phosphorylation of c-Jun NH_2_-terminal kinase (JNK), which induced cell apoptosis and ER stress mediated by the JNK inhibitor. Furthermore, AGE4 induced mitochondrial dysfunction by inducing the permeabilization of the mitochondrial membrane and ATP synthesis. Through in vitro and in vivo experiments, this study provides a new perspective on renal dysfunction with regard to the AGE4-induced RAGE /JNK signaling pathway, which leads to renal cell apoptosis via the imbalance of mitochondrial function and ER stress in kidney damage.

## 1. Introduction

Unlike type 1 diabetes, which is induced by an inherited disorder of insulin production in pancreatic islet β-cells [1], type 2 diabetes is caused by a variety of factors, including a lack of exercise, alcohol intake, genetic factors, and diet [2]. Diabetic nephropathy occurring in proximal renal epithelial cells is one of the main complications of diabetes [3]. Expression of receptors for advanced glycation end products (RAGE) is upregulated by abundant advanced glycation end products (AGEs) during diabetes-associated complications. The AGE–RAGE axis is involved in the onset of diseases such as Alzheimer’s disease, cancer, and osteoporosis [4,5,6]. According to recent studies, AGEs affect the glomerular filtration rate, resulting in chronic renal failure [7]. Alikhani et al. (2005) reported that N-ε-(carboxymethyl) lysine (CML)–collagen induced the apoptosis of fibroblasts, as well as increased caspase-3, -8, and -9 activity in an in vivo test [8]. Glyceraldehyde-derived AGEs induce apoptosis in human dermal fibroblasts by increasing reactive oxygen species (ROS) and activating the NLRP3 inflammasome [9]. Methylglyoxal (MGO), a major electrophilic dicarbonyl compound, is generated as a nonenzymatic breakdown product of a triosephosphate intermediate in the glycolytic process and has been linked to dicarbonyl stress, which leads to the development of AGEs and related cellular dysfunction [10,11]. We previously showed that the treatment of NRK-52E kidney cells with MGO-derived AGEs (AGE4) leads to an increase in the protein levels of matrix metalloproteinase-2 (MMP-2) and MMP-9 via AGE4–RAGE interactions [12]. Endoplasmic reticulum (ER) stress is an important mechanism that induces diabetes [13]. Moreover, mitochondrial functions include calcium homeostasis in cells, respiration, and biogenesis regulation through c-Jun NH2-terminal kinase (JNK) pathways [14,15,16], which are also closely related to cell apoptosis response pathways [17]. Although ER stress affects mitochondria directly or indirectly [18], the exact effects of AGE4 on the signaling pathways associated with ER stress and cell apoptosis are unknown. Therefore, we used in vitro and in vivo models to investigate the involvement of the AGE4–RAGE axis and specific signaling pathways that induce ER stress and mitochondrial dysfunction, which contribute to apoptosis.

## 2. Results

### 2.1. AGE4 Induces Cell Cycle Arrest and Apoptosis in HK-2 Cells

First, we investigated the effect of AGEs on HK-2 cell viability using the MTT assay. HK-2 cells were incubated at a concentration of 1 × 10^5^ cell/mL for 24 h. After incubation with AGEs for 24 h, values of the MTT assay were evaluated at concentrations of 100, 200, 300, and 400 g/mL. As seen in Figure 1a, cell viability of AGE-treated cells was not significantly different from that of unglycated control cells up to 200 g/mL of AGEs (AGE1, 100.6 ± 6.7%; AGE4, 99.4 ± 4.5%). However, in comparison with control cells, the viability of the cells treated with 300 or 400 μg/mL AGEs was found to be below 80%; thus, 200 μg/mL of AGEs were used in subsequent experiments. HK-2 cells were treated with AGE1 or AGE4 at 200 μg/mL for 24 h, stained with propidium (PI), and analyzed for cell cycle distribution. Figure 1b shows that AGE4 treatment decreased the percentage of HK-2 cells at the G1 phase, suggesting cell growth inhibition, and increased the percentage of sub-G1 phase and G2/M phase in the HK-2 cell cycle, indicating apoptosis progression and cell cycle arrest. Results of the HK-2 cell cycle analysis indicated that the sub-G1 distribution of the cell cycle increased in response to AGE4 (sub-G1: 5.2%, *p* < 0.05), whereas the number of cells in the G1 phase decreased in response to AGE4 (G1: 88.7%, *p* < 0.05) (Figure 1b). The results showed increased levels of HK-2 cells in the G2/M phase when treated with AGE4 (G2/M: 2%, *p* < 0.01). Cyclin D1 synthesis starts with G1, leading to a G1/S cell cycle transition, whereas cyclin B1 is present at low levels in normal tissue; nevertheless, cyclin D1 levels fall and cyclin B1 levels rise as a predictor of cell stress or death [19]. We determined the expression levels of cyclin D1 by Western blotting (Figure 1c). The AGE4 treatment lowered the protein level of cyclin D1, which is required for cell cycle growth. Furthermore, AGE4 treatment increased the mRNA level of cyclin B1, an important marker implicated in G2/M cell cycle arrest, as compared to the control (Figure 1d). These findings suggest that AGE4 can cause cell cycle arrest in HK-2 cells by inducing apoptosis.

Furthermore, AGE4 treatment (200 μg/mL) significantly (*p* < 0.05) increased not only the mRNA and protein levels of apoptosis markers such as Bax/Bcl2 and p53 but also the levels of a kidney damage marker, KIM-1, in HK-2 cells in comparison with BSA-treated control cells and AGE1-treated cells.

### 2.2. AGE4 Induces NF-κB, an Apoptosis-Promoting Transcription Factor

We analyzed the phosphorylation of c-Jun, JNK, ERK1/2, and p38 after AGE treatment for 24 h to determine which AGEs could regulate the MAPK pathways. We determined the cytosolic content of p-c-Jun/c-Jun, p-JNK/JNK, p-ERK/ERK, and p-p38/p38 in HK-2 cells (Figure 2a). JNK and c-Jun phosphorylation increased significantly (*p <* 0.05), but there was no difference in ERK1/2 and p38 activation by AGE4 treatment relative to BSA and AGE1 treatment, suggesting that AGE4 activates the c-Jun/JNK signaling pathway in HK-2 cells. The upregulation of NF-κB is important for activating the Nod-like receptor (NLR) family pyrin domain-containing 3 (NLRP3) inflammasome [20]. NF-κB is translocated from the cytoplasm to the nucleus as a transcription factor during ER stress and induces apoptosis in breast cancer cells [21]. We used Western blot and immunofluorescence (IF) assays to investigate alterations in NFB in the nucleus and apoptosis-related proteins in HK-2 cells at the protein level. Our results indicated that NF-κB, an important modulator of ER stress and apoptosis, was significantly (*p <* 0.05) increased in the cells treated with 200 μg/mL of AGEs compared to the BSA-treated cells.

Nuclear factor-kappa B (NF-κB) is a transcription factor that is involved in inflammatory and immune responses, as well as in the regulation of expression of many other genes related to cell apoptosis, death, proliferation, and differentiation. In mammalian cells, NF-B binds to the promoter region of target genes as a homodimer or heterodimer [22]. The NF-κB reporter (luc)-HEK293 cell line was designed for the study of NF-κB signal pathways. HEK293 cells were transfected with the luciferase reporter driven by the NF-κB promoter and then treated with 200 μg/mL of AGE4 treatment for 24 h (Appendix A). When compared to the control BSA treatment, AGE4 treatment significantly (*p <* 0.01) enhanced NF-κB promoter luciferase activity. As seen in Figure 2b,c, AGE4 treatment increased NF-κB in the nucleus (*p <* 0.05), whereas there was no upregulation of NF-κB following treatment with 10 μM SP600125, a JNK-specific inhibitor, for 3 h. Cytotoxicity of SP600125 was not identified at this concentration (Appendix A). Therefore, the activation of NF-κB by 200 μg/mL of AGE4 treatment can be regulated through the JNK signaling pathway (Figure 2d,e). Next, to investigate the role of the JNK signaling pathway in AGE4-induced ER stress and apoptosis, we preincubated the cells with 10 μM SP600125, a JNK-specific inhibitor, for 3 h. In HK-2 cells, the concentration of 200 μg/mL AGE4 treatment significantly (*p* < 0.05) induced p-JNK/JNK, p-c-Jun/c-Jun, CHOP, ATF4, GRP78, Bax, p53, and KIM-1, whereas SP600125 pretreatment inhibited JNK phosphorylation and the expression of ER stress and apoptosis markers (Figure 2f), suggesting that AGE4-induced ER stress and apoptosis can be controlled via the JNK signaling pathway. Three major ER stress sensors have been identified: inositol-requiring protein 1α (IRE1α), PKR-like ER kinase (PERK), and activating transcription factor 6 (ATF6) [23,24]. Under ER stress, these proteins initiate the unfolded protein response (UPR) signaling cascades to alleviate the burden of unfolded proteins. IRE1α is a transmembrane RNase involved in X-box-binding protein 1 (XBP1) expression. XBP1 is a major regulator of UPR, mediating adaptation to ER stress [25]. Figure 2g showed that the expression of XBP1 mRNA by 200 μg/mL of AGE4 treatment can be regulated through the JNK signaling pathway (*p <* 0.05) (Figure 2g).

### 2.3. RAGE–AGE4 Axis Induces ER Stress and Apoptosis in HK-2 Cells

RAGE, a specific receptor for AGEs on the cell surface, is known to mediate the MAPK pathway to induce inflammation, cell injury, and apoptosis [26,27]. There was a significant (*p <* 0.05) increase in RAGE protein and mRNA expression in AGE4-treated cells relative to BSA-treated control cells, but the expression was significantly (*p <* 0.05) suppressed by pretreatment with 10 μM FPS-ZM1 (a specific inhibitor of RAGE, Figure 3) or siRAGE transfection (Appendix A) in HK-2 cells in the presence of AGE4. ER stress proteins such as CHOP, ATF4, and GRP78 in AGE4-treated cells were significantly (*p <* 0.05) enhanced compared to those in the BSA-treated control cells. The protein levels of CHOP, ATF4, and GRP78 were significantly (*p <* 0.05) enhanced by AGE4 compared to those in the BSA-treated control cells, but FPS-ZM1-pretreated or siRAGE knockdown cells treated with AGE4 showed significantly (*p <* 0.05) reduced ER stress protein and mRNA levels (Figure 3, Appendix A). In contrast, the decrease in the level of Bcl-2, an anti-apoptosis marker protein, with AGE4 treatment was recovered with FPS-ZM1 pretreatment (Figure 3a, and 3d). Cytotoxicity by FPS-ZM1 was not identified (Appendix A). Furthermore, in HK-2 cells, AGE4 treatment at 200 g/mL significantly (*p <* 0.05) elevated p-JNK/JNK, whereas siRAGE knockdown reduced JNK phosphorylation (Appendix A), indicating that AGE4-induced JNK activation can be regulated via the AGE4–RAGE signaling pathway.

### 2.4. AGE4-Induced ER Stress Leads to Apoptosis

Salubrinal, an ER stress-specific inhibitor, was used to investigate the relationship between AGE4-induced ER stress and cell apoptosis. The cytotoxicity of 10 μM salubrinal for 3 h on HK-2 cells was not observed, and pretreatment with 10 μM salubrinal for 3 h decreased the level of ER stress proteins, such as in the 200 μg/mL of AGE4-treated group (Appendix A). Furthermore, the AGE4-induced increase in the protein expression levels of Bax/Bcl-2, p53, and KIM-1 was significantly (*p <* 0.05) decreased by the salubrinal pretreatment (Figure 4). These results imply that AGE4 is involved in ER stress-induced apoptosis in cells.

### 2.5. AGE4–RAGE Axis Induces ROS

The DCF-DA assay was used to determine whether 200 μg/mL of AGE4 treatment produced intracellular ROS. As a result, the percentage of ROS generation in the cells was 66.0 ± 7.6% with AGE1 treatment and 155.7 ± 7.6% with AGE4 treatment when compared to BSA treatment (Figure 5a). When the cells were pretreated with 10 μM of FPS-ZM1 for 3 h, the percentage of ROS generation was 81.8 ± 2.3% for AGE1 and 103.2 ± 2.3% for AGE4. It has been confirmed that the AGE4–RAGE-induced increase in the ROS levels is reduced by FPS-ZM1. However, the increase in ROS levels induced by AGE4 treatment was not inhibited by the pretreatment with 10 μM salubrinal for 3 h (89.7 ± 7.9% for AGE1, 153.5 ± 9.9% for AGE4) and 10 μM SP600125 for 3 h (81.4 ± 4.4% for AGE1, 134.5 ± 2.1% for AGE4). These results indicate that AGE4 produces ROS through RAGE in cells.

### 2.6. AGE4–RAGE Axis Signals Apoptosis via ER Stress

Cellular apoptosis was measured using an annexin V/PI staining kit. As shown in Figure 5b, the annexin V-FITC/PI-positive levels were increased by AGE4 treatment. The AGE4-treated cells exhibited an increased rate of cell apoptosis (BSA, 5.8 ± 0.8%; AGE1, 5.8 ± 2.8%; AGE4, 18.7 ± 2.1%), whereas this apoptotic response was significantly (*p <* 0.05) reduced by the pretreatment with 10 μM FPS-ZM1 for 3 h (BSA, 4.8 ± 0.42%; AGE1, 2.2 ± 0.67%; AGE4, 1.9 ± 0.6%) or 10 μM salubrinal for 3 h (BSA, 2.1 ± 1.55%; AGE1, 1.9 ± 1.1%; AGE4, 1.8 ± 0.25%), suggesting that AGE4 induces cellular apoptosis via ER stress mediated by RAGE–AGE4 interaction.

### 2.7. AGE4–RAGE Axis Induces Mitochondrial Dysfunction-Dependent ER Stress

Since the reduction of mitochondrial membrane potential (MMP) is a significant event in cellular apoptosis [28], MMP changes in cells treated with AGE were measured microscopically using JC-1 dye. AGE4 caused a remarkable increase in green fluorescence intensity compared to control BSA and AGE1, but green fluorescence intensity was not increased by treatment with AGE4 with FPS-ZM1 or salubrinal pretreatment (Figure 6a). In addition, MMP levels were decreased by AGE4 (46.56 ± 1.9%), but MMP levels increased upon treatment with FPS-ZM1 (85.19 ± 1.1%) or salubrinal (93.93 ± 1.6%) (Figure 6b). These results indicate that AGE4 can reduce MMP, leading to mitochondrial dysfunction in the cells and cellular apoptosis by damaging mitochondrial function. Next, as shown in Figure 6c, the level of ATP synthesis, one of the important functions of mitochondria, was reduced more by AGE4 treatment (74.59 ± 2%) than by AGE1 treatment (81.9 ± 0.04%), whereas the ATP synthesis level in the cells pretreated with FPS-ZM1 (AGE1, 91.54 ± 0.7%; AGE4, 99.05 ± 0.56%) or salubrinal (AGE1, 99.47 ± 2.2%; AGE4, 98.43 ± 0.66%) prior to AGE4 treatment was increased compared to that in cells treated with AGE4 only (Figure 6c). This suggests that ER stress induced by AGE4 is closely related to mitochondrial dysfunction, affecting MMP and ATP synthesis.

### 2.8. Effect of AGE4 on ER Stress and Apoptosis in an In Vivo Model

We have also demonstrated the effect of AGE4 on ER stress and mitochondrial dysfunction in the kidney using an in vivo C57BL/6N mouse model (4 weeks old, male). The mice were orally administered with 800 mg/kg bw of BSA, AGE1, or AGE4 for 3 weeks. The creatinine, glucose, BUN, and uric acid levels in C57BL/6N mice were (*p <* 0.05) significantly higher in the AGE4-fed group than in the BSA-fed group (Appendix A). In addition, it was found that the RAGE protein expression level was significantly (*p <* 0.05) higher in the AGE4-fed group than in the BSA- and AGE1-fed groups (Figure 7a). The protein expression levels of p-c-Jun/Jun and p-JNK/JNK were significantly (*p <* 0.05) induced by the oral administration of AGE4, whereas those of p-ERK and p-p38 failed to increase in the AGE4-fed group (Figure 7b). The protein level of cyclin D1 in the AGE4-fed group was significantly (*p <* 0.05) decreased compared with the BSA- and AGE1-fed groups (Figure 7c), whereas the mRNA level of cyclin B1 was considerably (*p <* 0.05) higher in the AGE4-fed group compared to the BSA- and AGE1-fed groups (Figure 7d). As with cell experiments, NF-κB protein expression in the nucleus of the mouse kidney tissue increased significantly (*p <* 0.05) compared to that in the cytoplasm in the AGE4-fed group (Figure 7e). The protein levels of CHOP, ATF4 (*p <* 0.05), GRP78, Bax/Bcl-2, and p53 were significantly (*p <* 0.05) high in the AGE4-fed group (Figure 7f). The mRNA results also show a significant (*p* < 0.05) increase in the AGE4-fed group compared to the BSA- and AGE1-fed groups (Figure 7g). These results indicate that AGE4 treatment also induces ER stress and can contribute to apoptosis in vivo in the mouse kidney. Based on the findings, we proposed that AGE4 treatment causes apoptosis in kidney cells (Figure 7h).

## 3. Discussion

Takeuchi et al. employed various sugar sources, including glucose, α-hydroxyaldehydes, and dicarbonyl compounds, to glycate proteins and classified AGEs into glucose-derived AGEs (AGE1); glyceraldehyde-derived AGEs (AGE2), glycolaldehyde-derived AGEs (AGE3), methylglyoxal-derived AGEs (AGE4), glyoxal-derived AGEs (AGE5), and 3-deoxyglucosone-derived AGEs (AGE6) [29,30]. Methylglyoxal, a reactive dicarbonyl produced during glucose metabolism, accumulates in diabetic patients [30], and its derived AGE4 is known to cause several diseases, including diabetes and cancer [11].

Our results clearly indicated that a significant decrease in cyclin D1 protein level was caused by AGE4. This decrease may result in an accumulation of cells in the sub-G1 and G1 phases. Cyclin D1 has been shown to be necessary for progression through the G1 process of the cell cycle in order to cause cell migration [31]. In addition, the activation of NF-κB by AGE4 treatment can be regulated through the JNK signaling pathway, according to our findings. RAGE–AGE4 axis-induced apoptosis is correlated with the induction of ER stress markers, such as CHOP, ATF4, and GRP78, via JNK pathway. In the previous study, AGE4 treatment induced RAGE-dependent cell inflammation in rat kidney epithelial cells (NRK-52E) through the ERK, JNK, and NF-κB pathways [12]. JNK and NF-κB are the main oncogenic signaling pathways linked to ER stress, which leads to nonalcoholic steatohepatitis and hepatocellular carcinoma [32].

The major source of ROS generation is the mitochondrial respiratory complexes, and the JNK pathway is involved in mitochondrial apoptosis [33,34]. AGEs play an important role in developing various diseases such as myocardial dysfunction and atherosclerosis [35,36], induce ROS through interaction with receptors, and regulate the JNK signal pathway [37,38]. In this study, AGE4 treatment of HK-2 human kidney cells induced ROS production through interacting with RAGE and activating the JNK signaling pathway, both of which play important roles in cellular apoptosis. Recent studies have shown that chronic inflammation induced by intracellular ROS is related to cell signaling in mitochondria and the development of metabolic diseases such as diabetes [39,40,41]. Cellular apoptosis is caused by oxidative stress-induced signal cascades related to cellular apoptosis involving Bcl-2 family proteins [42,43]. When Bcl-2 family proteins interact with the outer mitochondrial membrane, apoptosis proteins are activated [44,45]. On the other hand, due to physiological or pathological conditions, ER stress triggers inflammation, apoptosis, and cell death [46,47,48]. ER stress activates GRP78 and ATF4 through the phosphorylation of eIF2α and promotes the CHOP gene belonging to the C/EBP family, leading to cellular apoptosis [49].

## 4. Materials and Methods

### 4.1. Preparation of AGEs and Reagents

Fatty acid-free BSA fraction V (20 mg/mL) (Sigma-Aldrich, St. Louis, MO, USA) was incubated with 0.5 mM glucose (Sigma-Aldrich, St. Louis, MO, USA) or 20 mM methylglyoxal (Sigma-Aldrich, St. Louis, MO, USA) in 0.1 M potassium phosphate (Sigma-Aldrich, St. Louis, MO, USA) buffer containing 0.02% sodium azide (Sigma-Aldrich, St. Louis, MO, USA) and 1 mM diethylenetriaminepentaaceticacid (Sigma-Aldrich, St. Louis, MO, USA) in an O_2_ incubator (Vision Scientific Co., Daejeon, Republic of Korea) at 37 °C for 7 days. These glycated AGEs were classified as AGE1 (glucose-derived AGEs) and AGE4 (methylglyoxal-derived AGEs), respectively. The fluorescence of AGE formation was measured with a multi-microplate reader (excitation/emission = 370 nm/440 nm) (HIDEX, Turku, Finland). These AGEs were then dialyzed 24 h against potassium phosphate buffers to remove nonreactive sugars and other low-molecular-weight reactants. In the absence of sugar, nonmodified BSA was subjected to the same procedure.

JNK inhibitor SP600125, ER stress inhibitor salubrinal, and RAGE inhibitor FPS-ZM1 were purchased from Sigma-Aldrich (St. Louis, MO, USA). RAGE (sc-365154), ERK (sc-514302), p-ERK (sc-7383), c-Jun (sc-1694), p-c-Jun (sc-822), JNK (sc-571), p-JNK (sc-6254), p38 (sc-7972), p-p38 (sc-7973), NF-κB (sc-8008), CHOP (sc-7351), ATF4 (sc-390063), GRP78 (sc-166490), Bcl-2 (sc-56053) p53 (sc-126), cyclin D1 (sc-8396), and GAPDH (sc-32233) antibodies were obtained from Santa Cruz Biotechnology (Santa Cruz Biotechnology, Dallas, TX, USA). Bax (2772S) antibodies were obtained from Cell Signaling Technology (Danvers, MA, USA). KIM-1 (ab47634), Goat anti-mouse IgG H&L (Alexa Fluor 488) (ab150113) was obtained from Abcam (Cambridge, UK).

### 4.2. Cell Survival Rate and Oxidative Stress Generation

Human kidney proximal epithelial cells (HK-2) were cultured with RPMI-1640 (Gibco, Grand Island, NY, USA) containing 2 g of D-glucose, 2.383 g of HEPES, 2 g of sodium bicarbonate, 0.11 g of sodium pyruvate, 1% penicillin/streptomycin solution, and 10% fetal bovine serum (FBS) (*v*/*v*) (GE Healthcare Life Sciences, Chicago, IL, USA) in a 5% CO_2_ incubator at 37 °C. Cell viability was investigated using 3-(4,5-dimethyl thiazol-2-yl)-2,5-diphenyl tetrazolium bromide (MTT) assay. For cellular generation of ROS, HK-2 cells were seeded at 1 × 10⁵ cells/mL in culture 96-well black plate for 24 h and then treated with 10 μM 2′,7′-dichlorodihydrofluorescein diacetate (DCF-DA) (Sigma-Aldrich, St. Louis, MO, USA) with phenol-free medium for 30 min after washing with phosphate-buffered saline (PBS) twice. The AGEs were then treated in the cell and incubated in the dark, and the fluorescence was measured using a microplate reader (HIDEX, Turku, Finland).

### 4.3. In Vivo Animal Experiments

C57BL/6N mice (4 weeks of age, male) were purchase from Orientbio, Inc. (Seongnam, Gyeonggi-do, Korea). A standard pellet laboratory chow diet was provided while maintaining a 12 h light to 12 h dark cycle at 23 ± 2 °C with a relative humidity of 55% ± 5% under certain pathogen-free conditions (Young Bio, Seongnam, Gyeonggi-do, Korea). The mice were provided by the animal facility of Gyerim Experimental Animal Resource Center in Korea University. Before the feeding experiments, the animals were acclimatized for one week. The mice were divided into three groups (*n* = 6 for group). Mice were orally administered with 800 mg/kg body weight (bw) of BSA, AGE1, or AGE4 for 3 weeks. All animal experiments were conducted with the permission of the Korea University Institutional Animal Care and Use Committee and complied with the university’s Guidelines for the Care and Use of Laboratory Animals (KUIACUC-2018-6). No apparent sign of abnormality was observed during the experimental period.

### 4.4. Immunofluorescence (IF) Staining

HK-2 cells were seeded at a density of 1 × 10^5^ cells/mL on 35 mm dish plates for 24 h and treated with BSA and AGEs as described above. After 30 min fixation in 4% paraformaldehyde (Sigma-Aldrich, St. Louise, MO, USA) and being washed twice with PBS, HK-2 cells were permeabilized using 0.3% Triton X-100 in PBS and washed twice with PBS. Then, 1% BSA in PBS was added for 30 min for blocking. Cells were incubated with NF-кB antibody at 1:50 dilution in PBS overnight at 4°C (sc8008, Santa Cruz Biotechnology Inc., Santa Cruz, CA, USA). Cells were exposed to secondary goat anti-mouse IgG H&L (Alexa Fluor 488) (ab150113, Abcam, Cambridge, UK) for 1 h at room temperature in the dark. Nucleus counterstaining was achieved using DAPI (300 ng/mL in PBS, Sigma-Aldrich).

### 4.5. siRNA Knockdown Assay and Luciferase Assay

siRNA targeting RAGE and control siRAGE were purchased from Shanghai Gene Pharma Co., Ltd. (Shanghai, China). According to the manufacturer’s reagent protocol, 100 nM lipofectamine 2000 (Invitrogen, CA, USA) was used as a transfection reagent for HK-2 cells. After 24 h, total mRNA and protein were extracted to examine the RAGE knockdown level. The primer sequence was mentioned in a previous study [12], and the knockdown efficiency of mRNA and protein was determined based on a comparison with negative control siRNA.

Human embryonic kidney cells (HEK293 cells) were cultured with DMEM low glucose (Gibco, Grand Island, NY, USA) containing 2 g of sodium bicarbonate and 10% fetal bovine serum (FBS) (*v*/*v*) (GE Healthcare Life Sciences, Chicago, IL, USA) in a 5% CO_2_ incubator at 37 °C. HEK293 cells were plated at a density of 1 × 10^5^ cells/mL on 12-well plates for 24 h. After 50% cell confluency in 12-well plates, the pGL basic plasmid DNA and NF-κB promoter expressive plasmid DNA were transfected using lipofectamine 2000 for 24 h (Invitrogen, Carlsbad, CA, USA). AGEs were treated at a concentration of 200 μg/mL for 24 h. The growth medium was removed from the cultured cells, and phosphate-buffered saline (PBS) was gently added for washing. The luciferase activity was identified using Dual-Luciferase Reporter 1000 Assay System kit (Promega Corporation, Woods Hollow Rd, Fitchburg, MA, USA). Relative luciferase activity was measured with a filter luminescence microplate reader (HIDEX, Turku, Finland).

### 4.6. mRNA Preparation and Quantitative RT-PCR (q-RT-PCR) and Western Blotting

Whole mRNA was obtained using TRIzol solution (TAKARA Korea Biomedical Co, Seoul, Korea), and cDNA was manufactured using cDNA synthesis kit (LeGene Biosciences, San Diego, CA, USA). RAGE, CHOP, ATF4, GRP78, Bax, Bcl-2, p53, KIM-1, and GAPDH gene expression levels were analyzed using quantitative RT-PCR (q-RT-PCR). The primer sequence was obtained by Cosmo Genetech Co. (Seoul, Republic of Korea). The primer sequence is shown in Appendix A. The q-RT-PCR involved 40 cycles of denaturation at 95 °C, an extension for at 72 °C, and annealing at 54 °C. The cycle threshold (Ct) values were normalized to GAPDH as a control and expressed using the 2−ΔΔCt method. The q-RT-PCR amplifications were performed in a BioRad iQ5 cycler qRT-PCR detection system using the iQ SYBR Green Supermix (Bio-Rad Laboratories, Hercules, CA, USA).

For Western blotting, cells that had been treated with AGEs for 24 h were washed twice with cold PBS before treatment with RIPA lysis buffer (Elpis Biotech, Daejeon, Republic of Korea) containing protease inhibitors (5 μg/mL of aprotinin and 5 μg/mL of leupeptin), followed by separation of the supernatant using a centrifuge at 13,000 rpm for 20 min. To separate the nucleus and the cytoplasm, Thermo Scientific NE-PER nuclear and cytoplasmic extraction reagents (Thermo Fisher Scientific, Rockford, IL, USA) were used. The concentration of protein was quantified using a BCA protein assay kit (Thermo Fisher Scientific, Rockford, IL, USA). The protein isolated from cells was transferred from the 10% SDS-PAGE gel to a polyvinylidene difluoride (PVDF) membrane (Merck Millipore, Billerica, MA, USA). The PVDF membrane was incubated with 5% BSA or skim milk at 25 °C for 35 min and incubated overnight with primary antibody in a 4 °C experimental refrigerator at 25 °C for 45 min and then with a specific second antibody at 25 °C (Santa Cruz Biotechnology, Dallas, TX, USA). The protein bands were detected using Western BLoT ultra-sensitive HRP substrate (Takara Korea Biomedical Inc., Seoul, Korea) and enhanced chemiluminescence solution (Abclone, Seoul, Korea) and quantified using Image J software (National Institutes of Health, Bethesda, MD, USA).

### 4.7. 5′,6,6′-Tetrachloro-1,1′,3,3′-tetraethylbenzimidazolylcarbocyanine Iodide (JC-1) Staining and ATP Detection Assay

For measuring mitochondrial membrane potential, the cells were stained with 3 μM JC-1 (Koma Biotech, Seoul, Korea) for 30 min under dark condition and washed with PBS twice [50]. The mitochondrial membrane potential of fluorescence was measured using a confocal laser scanning microscope (Carl Zeiss AG, Jena, Germany) (green excitation/emission = 510/527 nm; red excitation/emission = 585/590 nm) and microplate reader (HIDEX, Turku, Finland) (green excitation/emission = 485/535 nm; red excitation/emission = 550/590 nm). Luminescence ATP detection assay system kit obtained from PerkinElmer Inc. (Waltham, MA, USA) was used to confirm the change in the composition of the mitochondrial ATP according to the company protocol. The ATP synthesis of mitochondria was measured using a microplate reader (HIDEX, Turku, Finland).

### 4.8. Cell Cycle Analysis and Annexin V/PI Staining

To examine cell apoptosis in response to AGE treatment, we used an annexin V/PI staining apoptosis reagent detection kit (BD Biosciences, San Diego, CA, USA). Cells were harvested, washed with cold PBS twice, resuspended in 1X working solution (10X binding buffer: 0.1 M HEPES, pH 7.4, 1.4 M NaCl, 25 mM CaCl_2_, diluted to 1X DW), and stained with fluorescein isothiocyanate (FITC)-labeled annexin V reagent in the dark for 20 min. Then, the cells were washed with 1X working solution and resuspended with PI apoptosis reagent. The percentage of cells was measured using flow cytometry (FACSCalibur, BD Bioscience Co., San Diego, CA, USA), and data were collected on a BD FACSCalibur flow cytometer using BD FACStation software system.

### 4.9. Statistical Analysis

Research data are expressed as mean ± standard deviations from at least three experiments. Statistical comparison between treatments was performed using one-way analysis of variance (ANOVA) and Tukey’s multiple tests. Student *t*-tests were used when comparing two groups. *P*-values of <0.05, < 0.01, and <0.001 were considered statistically significant.

## 5. Conclusions

In summary, the following points are suggested in this study: (1) RAGE–AGE4 can cause ER stress that leads to apoptosis in kidney cells. (2) The functions of the ER and mitochondria are interconnected, and the cellular mechanism of disorders is induced by the JNK signal pathway through AGE4–RAGE axis. Understanding the mechanism by which AGE4 induces ER stress and mitochondrial dysfunction during diabetic nephropathy will serve as a basis for a new therapeutic approach. Since AGE4, a type of AGE, causes cellular renal apoptosis, future research addressing the active component in AGE4 that causes diabetic nephropathy will aid in the delivery of treatments for diabetic complications.

## Figures and Tables

**Figure 1 ijms-22-06530-f001:**
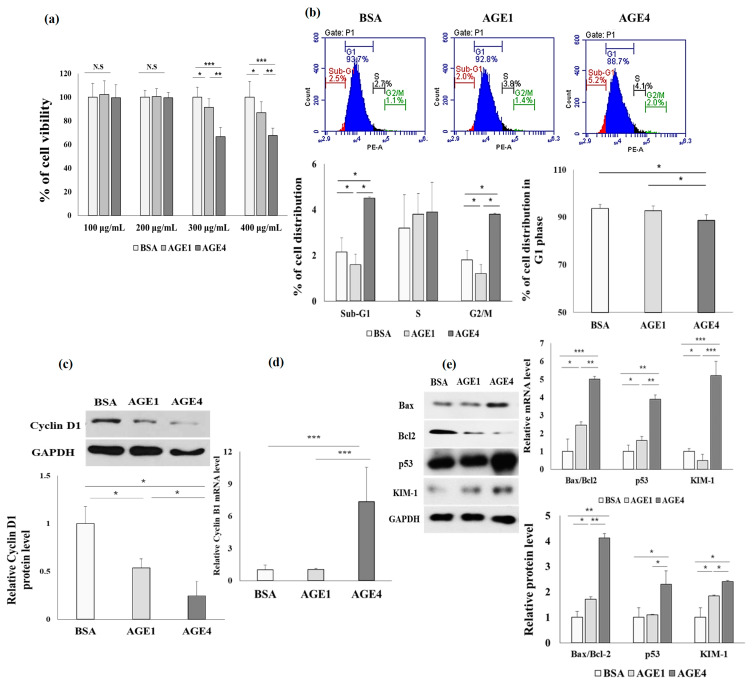
Effect of AGE4 treatment on HK-2 cells. (**a**) In HK-2 cells treated with AGE4, cell viability was determined using the MTT assay. (**b**) After staining with propidium iodide (PI), flow cytometry was used to analyze the cell cycle of HK-2 cells treated with 200 μg/mL of AGEs. The effect of AGE4 on the cell cycle was measured based on the distribution of cells in the sub-G1, G1, and G2/M phases. Red segment: sub-G1 phase of apoptosis; blue segment: G1 phase of cell development; and black segment: S phase of DNA synthesis. (**c**) The expression of cyclin D1 protein by AGE4 was investigated using Western blotting. Densitometric analysis was performed using ImageJ (NIH, Bethesda, MD, USA), and the ratio of the expression of the target protein to that of GAPDH was calculated. (**d**) The mRNA expression of cyclin B1 was detected by quantitative RT-PCR (q-RT-PCR). (**e**) The protein and mRNA expression levels of Bax/Bcl2, kidney injury molecule_1 (KIM-1), and p53 were detected by Western blotting and q-RT-PCR. Data are expressed as the mean ± standard deviation (SD) of three independent experiments. The groups were compared using a Student’s *t*-test, and the levels of statistical significance are indicated as follows: * *p* < 0.05, ** *p* < 0.01, and *** *p* < 0.001.

**Figure 2 ijms-22-06530-f002:**
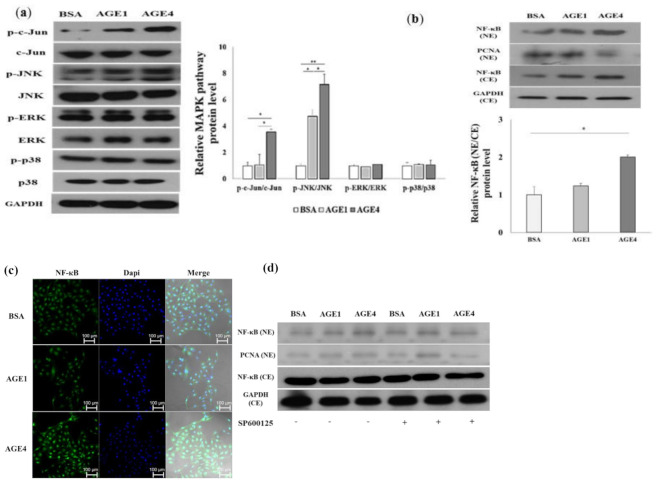
Effects of AGE4 on the ER stress and apoptosis through JNK signaling pathway in HK-2 cells. (**a**) With 200 μg/mL of AGE4 treatment for 24 h, the levels of phosphorylated c-Jun, JNK, ERK, and p38 protein were measured using Western blotting. (**b**,**c**) NF-κB translocation is represented by an increase in nucleus/cytoplasm ratio observed in the stained HK-2 cells shown from Western blot and immunofluorescence (IF) analysis. (**d**,**e**) NF-κB induction by AGE4 was inhibited by pretreatment with a JNK signal inhibitor (10 μM SP600125) shown from Western blot and IF analysis. Plus sign represents the presence of SP600125. Minus sign represents the absence of SP600125. (**f**) The protein expression levels of p-JNK, JNK, p-c-Jun, c-Jun, CHOP, ATF4, GRP78, Bax, Bcl2, p53, KIM-1, and GAPDH were detected using Western blot. Densitometric analysis was performed using ImageJ, and the ratio of the expression of the target protein to that of GAPDH was calculated. (**g**) The mRNA expression level of XBP1 was detected using q-RT-PCR. Data are expressed as the mean ± standard deviation (SD) of at least three independent experiments. The groups were compared using a Student’s *t*-test, and the levels of statistical significance are indicated as follows: * *p* < 0.05, ** *p* < 0.01, and *** *p* < 0.001.

**Figure 3 ijms-22-06530-f003:**
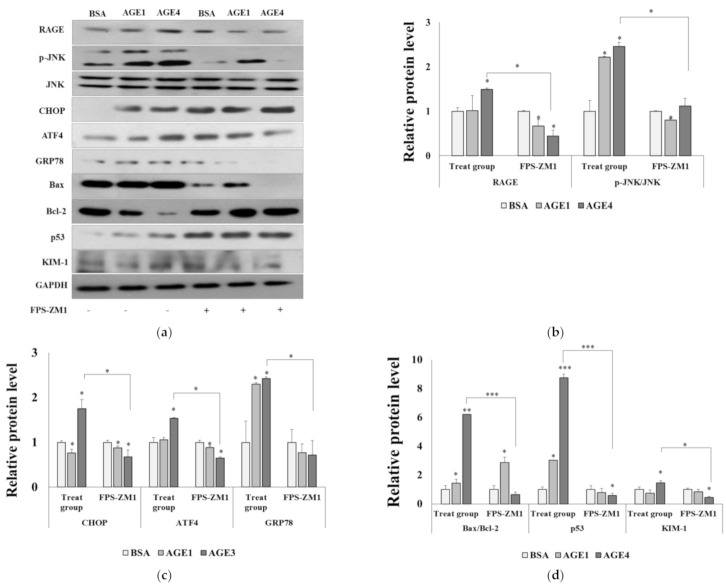
Effects of AGE4–RAGE axis on the ER stress and apoptosis in HK-2 cells. (**a**) After pretreatment with a selective RAGE inhibitor (10 M FPS-ZM1) and AGE4 (200 g/mL, 24 h), the protein expression levels of RAGE, p-JNK, JNK, CHOP, ATF4, GRP78, Bax/Bcl2, p53, and KIM-1 were determined by Western blotting. (**b**–**d**) Densitometric analysis was performed using ImageJ, and the ratio of the expression of the target protein to that of GAPDH was calculated. Plus sign represents the presence of FPS-ZM1. Minus sign represents the absence of FPS-ZM1. Data are expressed as the mean ± standard deviation (SD) of at least three independent experiments. The groups were compared using a Student’s *t*-test, and the levels of statistical significance are indicated as follows: * *p* < 0.05, ** *p* < 0.01, and *** *p* < 0.001.

**Figure 4 ijms-22-06530-f004:**
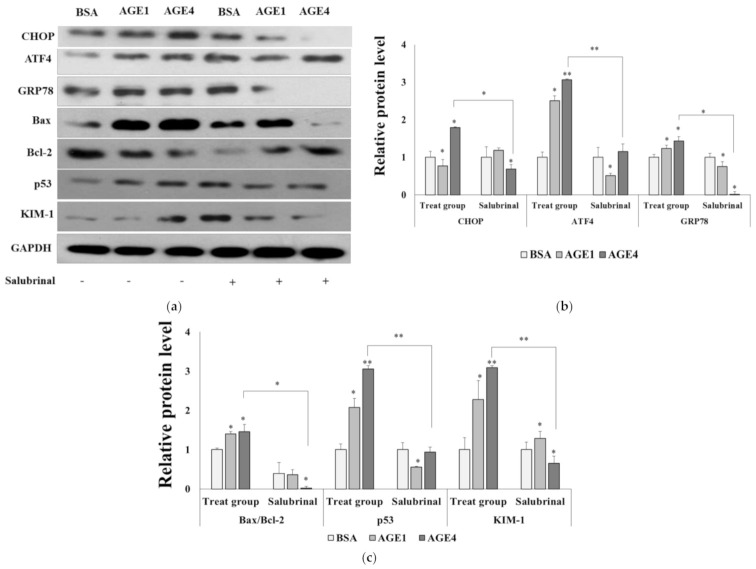
Effects of AGE4 on the ER stress and apoptosis in HK-2 cells. (**a**) After pretreatment with 10 μM, salubrinal and AGE4 (200 μg/mL, 24 h), the protein expression levels of CHOP, ATF4, GRP78, Bax, Bcl2, p53, and KIM-1 were determined by Western blotting. (**b**,**c**) Densitometric analysis was performed using ImageJ, and the ratio of the expression of the target protein to that of GAPDH was calculated. Plus sign represents the presence of salubrinal. Minus sign represents the absence of salubrinal. Data are expressed as the mean ± standard deviation (SD) of at least three independent experiments. The groups were compared using a Student’s *t*-test, and the levels of statistical significance are indicated as follows: * *p* < 0.05, and ** *p* < 0.01.

**Figure 5 ijms-22-06530-f005:**
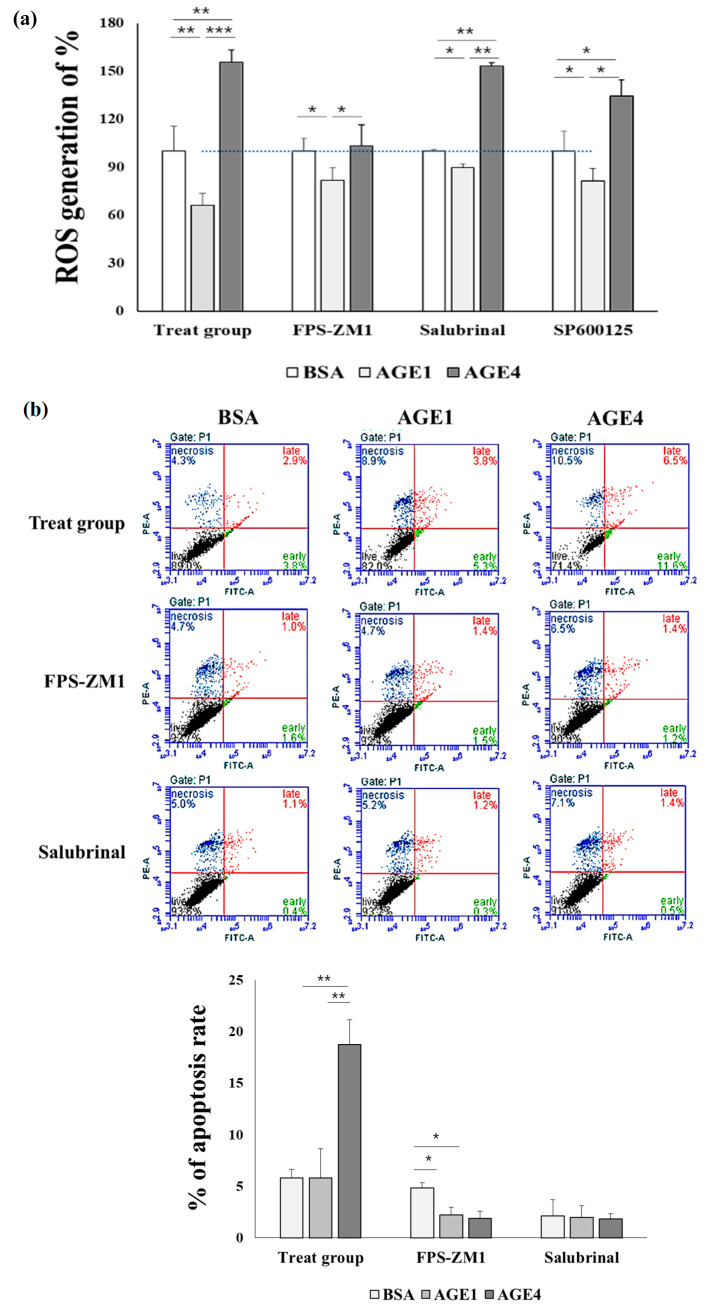
Effects of AGE4–RAGE axis on the reactive oxygen species (ROS) and apoptosis in HK-2 cells. HK-2 cells were pretreated for 3 h with 10 μM FPS-ZM1 or 10 μM salubrinal before being treated with 200 g/mL of AGE4. (**a**) The generation of ROS in response to AGE4 was assessed by DCF-DA assay. Black segment: live cells, lower-left quadrant; green segment: early apoptosis cells, lower-right quadrant; red segment: late apoptosis cells, upper-right; and blue segment: necrosis cells, upper-left. (**b**) Necrosis and early and late apoptotic cells were detected by the annexin V/PI staining and flow cytometry. Each of these experiments was conducted independently more than three times. Data are expressed as the mean ± standard deviation (SD) of at least three independent experiments. * *p* < 0.05, ** *p* < 0.01, and *** *p* < 0.001; groups were compared by *t*-test.

**Figure 6 ijms-22-06530-f006:**
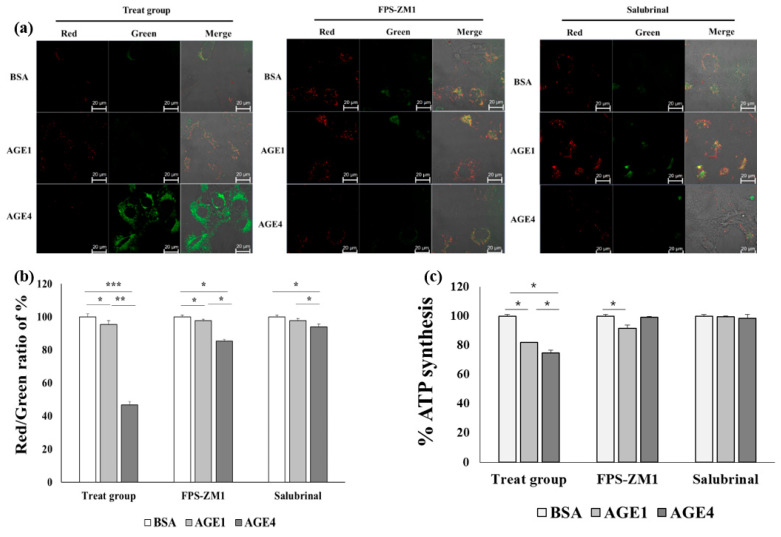
Effects of AGE4–RAGE axis on the mitochondrial dysfunction in HK-2 cells. (**a**) The mitochondrial membrane potential (MMP) was measured by confocal microscope. A microscope was used to amplify each image to magnification of 100. (**b**) MMP was determined using a fluorometric assay on a multi-microplate reader. (**c**) The extent of ATP synthesis was determined using an ATP detection assay. The experiments were conducted independently more than three times. * *p <* 0.05, ** *p <* 0.01, and *** *p <* 0.001; groups were compared by *t*-test.

**Figure 7 ijms-22-06530-f007:**
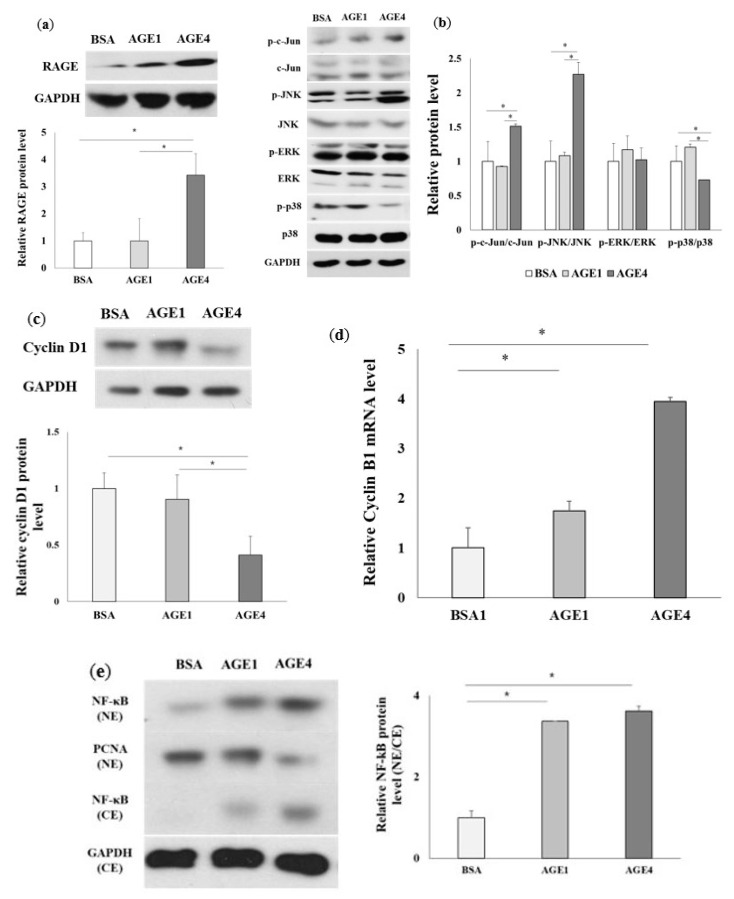
Effects of AGE4 on kidney cell apoptosis in an animal model. The kidney tissue was isolated, and the expression of each marker was determined using Western blot and q-RT-PCR. (**a**–**c**) The protein expression levels of RAGE, c-Jun, JNK, ERK, P38, and cyclin D1 were detected by Western blot. (**d**) NF-κB translocation is represented by an increase in nucleus/cytoplasm ratio of its staining in mouse kidneys. (**e**) The protein levels of CHOP, GRP78, ATF4, Bax/Bcl-2, and p53 were detected by Western blot. Densitometric analysis was performed using ImageJ, and the target protein to GAPDH ratio was calculated. (**f**) The mRNA expression levels of RAGE, CHOP, ATF4, GRP78, Bax/ Bcl-2, and p53 were measured by q-RT-PCR. (**g**) A schematic diagram of the influence of the AGE4-induced signal pathway on the endoplasmic reticulum (ER) and mitochondria in the kidney. (**h**) AGE4: methylglyoxal-derived AGEs, RAGE: receptors for advanced glycation end products, ROS: reactive oxygen species. * *p* < 0.05, ** *p* < 0.01, and *** *p* < 0.001; groups were compared by *t*-test.

## Data Availability

Data available upon reasonable request to the authors.

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
