# Peer review of "Methylglyoxal-Derived Advanced Glycation End Product (AGE4)-Induced Apoptosis Leads to Mitochondrial Dysfunction and Endoplasmic Reticulum Stress through the RAGE/JNK Pathway in Kidney Cells"

_ijms, 2021, doi:10.3390/ijms22126530_

Round 1
Reviewer 1 Report
The manuscript named Methylglyoxal-derived advanced glycation end product (AGE4)-induced apoptosis leads to mitochondrial dysfunction and endoplasmic reticulum stress through the RAGE/JNK pathway in kidney cells is presented for peer review. Type 2 diabetes is a major health concern worldwide. A variety of factors caused it, including the sedentary lifestyle, alcohol intake, genetic factors, and diet. Expression of receptors for advanced glycation end products (RAGE) is upregulated by abundant advanced glycation end products (AGEs) during diabetes-associated complications. The AGE-RAGE axis is involved in the onset of diseases such as Alzheimer's disease, cancer, and osteoporosis. Authors investigate the involvement of the AGE4-RAGE axis and specific signaling pathways that induce ER stress and mitochondrial dysfunction, which contribute to apoptosis. Methods are sound and the reference list is updated. I have several suggestions to you.
First, please explain the choice of the drug concentration for AGE4 and AGE1
Second, Fig.1 needs formatting.
Third, I suggest probing cyclinB1 for G2/M transition in addition to cyclin D.
Next, I advise using the siRNA to inhibit JNK signaling (both JNK1 and JNK 2 isoforms) after AGE treatment to further prove SP600125 action.
Additional suggestion deals with transcritional activity regulation. I think using NF-kB reporter system would improve the paper.
Please provide size bars in Fig.6A lines 237-254.
Reviewer 2 Report
The study entitled ‘Methylglyoxal-derived advanced glycation end product (AGE4)-induced apoptosis leads to mitochondrial dysfunction and endoplasmic reticulum stress through the RAGE/JNK pathway in kidney cells’ by Jeong and Lee aims to reveal a possible link between methylglyoxal-derived advanced glycation end-products-induced cell death and mitochondrila dysfunction/endoplasmic reticulum stress in kidney cells. The authors suggest that AGE4 induces RAGE/JNK signaling leading to renal cell apoptosis via the imbalance of mitochondrial function and ER stress.
For this reviewer, the main strength of this study that the in vitro data are supported by animal experiments that counterbalance the limitations of the cell culture experiments.
Main issues
- The main limitation for this reviewer that untreated cells are not included in the cell culture experiments that makes difficult the perform a real comparison of the different groups; the data on AGEs are only compared to BSA-data.
- Fig 1A: Does AGE4-induced cell death (300 µg/mL) statistically significant? An appropriate statistical test should be performed. Is AGE-induced cell dose-dependent? Did the authors try higher concentrations of AGE?
- Fig 2D: What was the concentration of AGEs in these experiments? 200 or 300 µg/mL? It is an important issue throughout the whole in vitro experiments that the AGE concentration used is not included either in main text or the figure legend.
- Fig 2 B,C: NFΚB nuclear translocation should be followed by immunofluorescence.
- Fig 2D: in Fig 2A, the authors showed that AGEs induce c-jun phosphorylation. Why the authors did not measure this in Fig 2D to support the effect of JNK inhibitor SP600125? The authors claim in line 140-141 that ‘SP600125 pretreatment inhibited JNK phosphorylation and the expression of ER stress and apoptosis markers’. It is supported by CHOP blot but not by ATF4. Grp78 is difficult to evaluate. Given that JNK can be activated by the IRE1α arm of ER stress, I’d suggest to measure XBP1 activation by PCR or immunoblot in addition to these data.
- In line 154, the authors wrote that ‘FPS-ZM1-treated cells or siRAGE knockdown cells treated with AGE4 showed significantly reduced ER stress protein and mRNA levels’, however, this is not supported well by measuring CHOP in Fig 4. I’d recommend to provide an image that better supports this.
- Figure 5. Concentrations of AGEs and inhibitors are missing. Is the title of 2.6. appropriate? RAGE-AGE4 axis induces apoptosis-dependent ER stress? Is it correct? In addition, I’d recommend to test other ER stress inhibitors, such as 4-Phenylbutyric acid or Ursodeoxycholic acid that would considerably support this finding.
Round 2
Reviewer 1 Report
Authors addressed all my questions. Thank you.
Reviewer 2 Report
The authors have satisfactorily addressed my concerns I recommend the manuscript for publication.